# Doxycycline Ameliorates the Severity of Experimental Proliferative Vitreoretinopathy in Mice

**DOI:** 10.3390/ijms222111670

**Published:** 2021-10-28

**Authors:** Shun-Hua Chen, Yu-Jheng Lin, Li-Chiu Wang, Hsien-Yang Tsai, Chang-Hao Yang, Yu-Ti Teng, Sheng-Min Hsu

**Affiliations:** 1Department of Microbiology and Immunology, College of Medicine, National Cheng Kung University, Tainan 701, Taiwan; shunhua@mail.ncku.edu.tw (S.-H.C.); ca130705@gmail.com (Y.-J.L.); 2School of Medicine, I-Shou University, Kaohsiung 824, Taiwan; statolish@isu.edu.tw; 3Department of Ophthalmology, Tzu Chi Hospital, Taichung 427, Taiwan; tc1512901@tzuchi.com.tw; 4Department of Ophthalmology, National Taiwan University Hospital, College of Medicine, National Taiwan University, Taipei 100, Taiwan; chyangoph@ntu.edu.tw; 5Department of Ophthalmology, National Cheng Kung University Hospital, College of Medicine, National Cheng Kung University, Tainan 704, Taiwan; andyteng1026@gmail.com

**Keywords:** doxycycline, proliferative vitreoretinopathy (PVR), rhegmatogenous retinal detachment (RRD)

## Abstract

After successful surgeries for patients with rhegmatogenous retinal detachment, the most common cause of retinal redetachment is proliferative vitreoretinopathy (PVR), which causes severe vision impairment and even blindness worldwide. Until now, the major treatment for PVR is surgical removal of the epiretinal membrane, while effective treatment to prevent PVR is still unavailable. Therefore, we investigated the potential of doxycycline, an antibiotic in the tetracycline class, to treat PVR using a mouse model. We used the human retinal pigment epithelial cell line, ARPE-19, for in vitro and in vivo studies to test doxycycline for PVR treatment. We found that doxycycline suppressed the migration, proliferation, and contraction of ARPE-19 cells with reduced p38 MAPK activation and total MMP activity. Intravitreal doxycycline and topical tetracycline treatment significantly ameliorated the PVR severity induced by ARPE-19 cells in mice. PVR increased the expression of MMP-9 and IL-4 and p38 MAPK phosphorylation and modestly decreased IL-10. These effects were reversed by doxycycline and tetracycline treatment in the mouse retina. These results suggest that doxycycline will be a potential treatment for PVR in the future.

## 1. Introduction

Rhegmatogenous retinal detachment (RRD) is a severe and potential sight-threatening disease. The annual incidence of RRD is between 5.4 and 18.2 per 100,000 people [1,2,3,4]. The primary surgical reattachment rate of RRD is from 75% to 90%, but around 10% of initially successful cases result in retinal redetachment due to proliferative vitreoretinopathy (PVR) [5,6,7,8], which is the most common cause of failed repair of RRD [1,5,9]. PVR originates at the time of retinal break, followed by the migration of retinal pigment epithelial (RPE) cells from the normal location to the vitreous cavity. Then, the formation of scar-like fibrovascular membranes occurs, which induce retinal redetachment as they contract [9]. The current treatment options for PVR are various surgical procedures, including vitrectomy, membrane peeling, relaxing retinotomies, the use of liquid perfluorocarbons, and internal tamponade with gas or silicon oil [1]. However, the surgical success rate for PVR is approximately 60–75% at 6 months, and more than 25% of initially successful cases result in retinal redetachment due to recurrent retinal traction [1]. PVR is the result of the growth and contraction of cellular membranes within the hyaloid and retinal surface. These membranes exert traction and may cause tractional retinal detachment. The principal cells involved in PVR are RPE cells, retinal glial cells, and macrophages [10]. Since the visual results of PVR are unsatisfactory even after surgical treatment, many efforts have been directed towards pharmacological prevention of cellular proliferation and membrane contraction for PVR.

Although PVR is currently primarily managed surgically, it is virtually impossible for vitrectomy to prevent cell adhesion and pathological change. Therefore, control of the biological processes involved in cell proliferation and wound healing would improve the success rate of surgery for PVR [1]. The results of clinical trials assessing the efficacy of 5-fluorouracil and low-molecular-weight heparin, daunorubicin, corticosteroids, colchicine and 13-cis-retinoic acid are mixed [5,11,12,13,14,15,16,17]. Intravitreal (IVI) methotrexate injection has been extensively studied for PVR prevention recently. Although some promising results have been reported, the effect has not yet been proven in the clinical trials [18,19,20,21]. Many drugs, such as corticosteroids, retinoids, and glucosamine, might be useful for inhibiting PVR development at specific stages [1,22,23,24]. However, more work is necessary to identify optimal adjuvant therapies for the management of PVR.

In PVR progression, the levels of several cytokines, chemokines, growth factors, and enzymes are elevated in the pathological vitreous epiretinal membranes (ERMs) of patients. Cytokines include interleukins (IL)-1β, -4, -6, and -8; tumor necrosis factor (TNF)-α; and interferon (IFN)-γ, as well as chemokines, including monocyte chemotactic proteins (MCP)-1 and 2; CC chemokine ligands (CCL)-19, -22, -23, and -27; and C-X-C motif chemokines (CXCL)-6, -9, -10, and -12, which are involved in inflammation, wound healing, and several cellular processes [25,26,27,28]. The enzymes involved in PVR are matrix metalloproteinases (MMP)-1, -2, -3, -8, and -9 and tissue inhibitor of metalloproteinase (TIMP)-1 [29,30,31].

Doxycycline, a Food and Drug Administration-approved antibiotic, is one of the potent and well-tolerated drugs in the tetracycline family. It can act as an MMP inhibitor to decrease cell proliferation, migration, adhesion, and contraction by suppressing the activities and expression of MMP-1, -2, -3, -8, or -9 [32,33,34,35,36,37]. It can also act as a p38 MAPK or NF-κB inhibitor to reduce the expression of cytokines, such as IL-1β, IL-6, IL-8, and TNF-α in epithelial cells [38,39]. Doxycycline has been used in clinical trials for patients with severe nonproliferative diabetic retinopathy [40,41]. However, doxycycline has not been used as a treatment for PVR. In this study, we show that doxycycline not only decreased cell migration, proliferation, and contraction, but also suppressed the activity levels of MMPs and p38 MAPK in the human RPE cell line, ARPE-19. A previous study showed that topical tetracycline ointment can produce drug levels in the aqueous humor [42]. Therefore, we also used a combination of IVI doxycycline and topical tetracycline treatment in the PVR mouse model. We found that IVI doxycycline and topical tetracycline treatment reduced the severity of experimental PVR in mice.

## 2. Results

### 2.1. Doxycycline Decreases the Migration, Proliferation, and Contraction of ARPE-19 Cells

During PVR progression, RPE cells can migrate from the retina to the vitreous body and can proliferate to form the ERM, which can cause retinal detachment because of its contractile ability [43]. To assess the potential of doxycycline for PVR treatment, we first examined whether doxycycline can inhibit cell migration using the human RPE cell line, ARPE-19. Doxycycline (40 μM, but not 20 or 10 μM) inhibited the cell migration 24, 48, and 72 h after drug treatment when compared with the untreated cells (Figure 1A). We next evaluated whether doxycycline can reduce the proliferation of ARPE-19 cells, which proliferated after culture (Figure 1B). Doxycycline (50 μM, but not 40, 20, or 10 μM) reduced the cell proliferation 48 and 72 h, but not 24 h, after drug treatment. We also examined whether doxycycline can decrease the contraction of ARPE-19 cells by seeding cells in collagen gels. After 24 h, ARPE-19 cells caused collagen gel contraction by 40 to 60% (Figure 1C). Doxycycline (10, 20, and 40 μM) decreased the cell contraction in a dose-dependent manner 24 to 72 h after drug treatment with the half-maximal inhibitory concentration (IC_50_) of 10 μM after drug treatment for 48 h. Collectively, we found that treatments with 10, 40, and 50 μM of doxycycline for 24, 24, and 48 h inhibited the contraction, migration, and proliferation of APRE-19 cells, respectively.

### 2.2. Doxycycline Diminishes the Total MMP Activity and p38 MAPK Activation of ARPE-19 Cells

MMPs are enzymes detected in the vitreous body of PVR patients, which can promote cell migration, proliferation, and contraction [29,30,44]. Doxycycline has been shown to suppress MMP activity indirectly by increasing TIMPs, which inhibits MMP activity [32]; it can also act directly [37]. We determined the effect of doxycycline on the total MMP activity of ARPE-19 cells and found that doxycycline at the concentration of 10 μM suppressed the total MMP activity 24, 48, and 72 h after drug treatment (Figure 2A). We next tested the effects of doxycycline on MMP and TIMP levels. We detected MMP-1, -2, -3, -9, and -13 as well as TIMP-1, -2, and -4 in ARPE-19 cells and found comparable levels of these proteins in the cells treated with or without doxycycline using a human antibody array (Appendix A). We also performed Western blotting to determine the effects of doxycycline on MMP and TIMP levels. Doxycycline at 10 and 40 μM failed to reduce the levels of MMP-3, while doxycycline at 40 μM, but not at 10 μM, slightly decreased MMP-8 and MMP-9 (Appendix A). Accordingly, doxycycline might inhibit MMP activity directly.

Doxycycline has been shown to inhibit TGF-β-induced MMP-9 production and activity through the p38 MAPK signaling pathway in primary human corneal epithelial cells [44]. We, therefore, examined the effect of doxycycline on TGF-β-induced p38 MAPK activation (phosphorylation) using Western blotting. TGF-β increased the level of phosphorylated p38 MAPK (p-p38 MAPK), while doxycycline (40 μM) reduced the levels of p-p38 MAPK without affecting the levels of total p38 MAPK in ARPE-19 cells (Figure 2B–D), showing the capacity of doxycycline to reduce p38 MAPK activation in ARPE-19 cells.

### 2.3. Doxycycline Reduces PVR Severity in Mice

We evaluated the effects of doxycycline on PVR in mice. We established the PVR disease model in mice by injecting ARPE-19 cells into the vitreous body, as previously described [45]. After testing several cell doses, we found that IVI injection of 1.6 × 10^4^ cells/vitreous body successfully induced PVR in mice (Appendix A). We then tested the potential of doxycycline to reduce PVR in vivo using mice intravitreally injected with ARPE-19 cells in saline with or without doxycycline. Additionally, mice were treated with or without tetracycline ointment topically on the cornea once every two days for 14 days. We examined the PVR progression in the mouse fundus using a slit lamp every week, starting from the day of induction (day 0). We tested several doses of doxycycline and found that IVI injection of doxycycline (5 μg in 2 μL; 4.8 mM) alone reduced the PVR disease score 21 days after PVR induction when compared to the saline-treated group (Figure 3A). IVI injection of doxycycline plus topical tetracycline treatment further decreased the PVR disease score in mice. However, topical tetracycline treatment alone failed to improve PVR. In mouse fundus images, we found ERM, retinal detachment, and inflammation 21 days after PVR induction (Figure 3B). Doxycycline and tetracycline treatment decreased the PVR pathology, ERM, and retinal detachment in the mouse fundus. We also harvested mouse eye samples 21 days after PVR induction to examine the retinas using hematoxylin and eosin (H&E) staining and found that the cellular arrangement of the retinas with PVR was disrupted (Figure 3C). Doxycycline and tetracycline treatment improved the retinal morphology of PVR mice.

### 2.4. Doxycycline and Tetracycline Treatment Decreases MMP-9 Expression and p38 MAPK Phosphorylation in the Retina and Vitreous Body of PVR Mice

We examined the effects of doxycycline and tetracycline treatment on the expression of MMP-9, which is highly expressed in the vitreous of patients with PVR [30]. The small size of the mouse eyes precluded the collection of vitreous samples. We, therefore, collected the mouse eyes 21 days after PVR induction to detect MMP-9 by immunofluorescence staining. MMP-9 was rarely detected in the retina or vitreous of naïve mice (Figure 4). Abundant MMP-9 was detected in the mouse retina and vitreous after PVR induction. Doxycycline and tetracycline treatment greatly reduced MMP-9 expression in the mouse retina and vitreous after PVR induction.

We further searched the signal molecule, which can increase MMP 9 expression. After activation (phosphorylation), p38 MAPK can increase MMP-9 production [46]. We, therefore, tested the effects of doxycycline and tetracycline treatment on p38 MAPK phosphorylation by collecting mouse eyes 21 days after PVR induction to detect p-p38 MAPK by immunofluorescence staining. In the retina and vitreous of naïve mice, p-p38 MAPK was rarely detected (Figure 5A). After PVR induction, p-p38 MAPK was detected in the retina, and especially in the vitreous of mice. Doxycycline and tetracycline treatment reduced p-p38 MAPK in the mouse retina and vitreous of PVR mice. We also harvested the mouse retina samples 21 days after induction and examined the levels of p38 MAPK using Western blotting. The mouse retinas showed that PVR induction increased the ratio of p-p38 MAPK/p38 MAPK and that doxycycline and tetracycline treatment reduced the ratio of p-p38 MAPK/p38 MAPK after PVR induction (Figure 5B–D). The retinal levels of p38 MAPK in naïve mice and PVR mice treated with saline or doxycycline and tetracycline were comparable. Accordingly, in the retina, PVR induces p38 MAPK activation, while doxycycline plus tetracycline treatment decreases p38 MAPK activation in PVR mice. Additional Western blotting results showed that doxycycline and tetracycline treatment failed to significantly affect the activation (phosphorylation) of other signal molecules, ERK and JNK, which can increase MMP 9 expression, in the retina of PVR mice.

Doxycycline is an inhibitor of NF-κB [39], which can regulate cell proliferation [47], migration, and contraction once it is activated. NF-κB activation can be demonstrated by the translocation of its component p65 to the nucleus. Therefore, to examine the effect of doxycycline on NF-κB activation, the nuclear and cytoplasmic proteins of ARPE-19 cells were extracted after treatment with or without doxycycline for analysis of the NF-κB subunit, p65, by Western blotting. The results showed that doxycycline (10 and 40 μM) reduced the levels of p65 translocation into nucleus in a dose-dependent manner 48 h after drug treatment (Appendix A), suggesting that doxycycline reduces NF-κB activation of ARPE-19 cells.

### 2.5. Doxycycline and Tetracycline Treatment Decreases IL-4 and CXCL9 but Increases IL-10 in the Retina of PVR Mice

During PVR progression, several cytokines and chemokines are detected in the pathological vitreous or ERM [28]. We, therefore, harvested the mouse retina samples 21 days after PVR induction to monitor cytokines and chemokines using the Luminex assay. IL-4 and CXCL9 are increased in the vitreous of patients with PVR [28]. Our results showed that IL-4 and CXCL9 levels were also elevated in the mouse retina after PVR induction (Figure 6A,B). Doxycycline and tetracycline treatment reduced IL-4 and CXCL9 levels in the mouse retina 21 days after PVR induction. IL-10 has been shown to inhibit MMP-9 production [48]. Our results showed that doxycycline and tetracycline treatment increased the IL-10 level in the mouse retina after PVR induction (Figure 6C). Additional results showed that doxycycline and tetracycline treatment failed to affect the levels of other cytokines (IFN-γ, IL-6, and TNF-α), chemokines (CXCL10 and MCP-1), or vascular endothelial growth factor.

## 3. Discussion

Our in vitro study demonstrated that doxycycline can reduce the migration, proliferation, and contraction of ARPE-19 cells with reduced total MMP and p38 MAPK activities. Furthermore, mouse studies showed that IVI doxycycline plus topical tetracycline treatment could effectively ameliorate the severity of PVR with decreased levels of IL-4 and MMP-9, which can be downregulated by p38 MAPK phosphorylation and increased by IL-10.

PVR development arises from several cell types, especially RPE cells, which migrate, proliferate, and form an ERM in patients with RRD. The contractile ability of the ERM causes the tractional retinal detachment and induces PVR. Therefore, blocking the migration, proliferation, and contraction of RPE cells will be helpful and important in reducing PVR formation. Our in vitro studies demonstrated that doxycycline significantly reduces the migration, proliferation, and contraction of ARPE-19 cells, suggesting that doxycycline could potentially block PVR.

In animal studies, we chose ARPE-19 cell-induced PVR mice as the animal model and intravitreally injected APRE-19 cells with saline or doxycycline to test whether doxycycline could reduce the severity of PVR in mice. In the ARPE-19 cell-induced PVR mice, lesions with ERM formation, inflammation, and tractional retinal detachment could be observed in a manner similar to those observed in PVR patients. We found that IVI doxycycline significantly reduces PVR in mice. Additional topical tetracycline treatment further decreased the severity of PVR, although topical tetracycline treatment alone failed to reduce PVR formation. Therefore, IVI injection of doxycycline plays a vital role in reducing PVR development. Previous studies have shown good ocular penetration after systemic doxycycline or tetracycline treatment in animals [49,50,51]. Therefore, we can anticipate the effects of oral doxycycline for PVR treatment in human patients.

Previous studies showed elevated MMP-1, -2, -3, -8, -9, and TIMP-1 levels in the vitreous of PVR patients, suggesting a possible role for MMPs in PVR pathophysiology [29,30]. MMPs can also induce epithelial–mesenchymal transition (EMT) in breast cancer [52], while EMT of RPE cells has been demonstrated to be an important mechanism for PVR formation [43,45]. Doxycycline, the antibiotic in the tetracycline class, can also inhibit the activity of some MMPs in skin, cartilage, and corneal collagenase [32]. Therefore, in this study, we tested and found that doxycycline decreases the migration, proliferation, and contraction of ARPE-19 cells with a reduced level of total MMP activity. Additionally, in the PVR animal model, we also found that doxycycline can ameliorate PVR with reduced MMP-9 expression in the vitreous of mice, as demonstrated by immunofluorescence staining. However, since the vitreous of the mice is too hard to collect, we cannot check the expression of MMP-9 by Western blotting. Previous studies found that doxycycline suppresses microglial activation by inhibiting p38 MAPK and NF-κB signaling pathways [53]. We have also checked both pathways using in vitro and in vivo studies. Our in vitro results showed that doxycycline inhibits the total MMP activity by suppressing the activation of both p38 MAPK and NF-κB. However, our PVR mouse results showed that only the activation of p38 MAPK, but not of NF-κB, is suppressed. A previous report showed that in rat primary astrocytes, p38 MAPK phosphorylation could elevate the level of MMP-9 [46]. Therefore, doxycycline could suppress MMP-9 expression by reducing p38 MAPK phosphorylation to ameliorate PVR in our mouse PVR model.

T helper (Th) cells have been reported to be involved in PVR development and exist in the ERM of PVR patients [54]. Naïve Th cells can be stimulated by IL-4, which then differentiate to Th2 cells, promoting tissue fibrosis [55,56]. We found that doxycycline and tetracycline treatment can eliminate the ERM formation in mice with reduced levels of IL-4 in the retina. Human eyes with PVR also showed higher levels of CXCL9 [28], as in our mouse PVR model. On the other hand, IL-10 is an anti-inflammatory cytokine that was shown to decrease MMP-9 activity in a previous report [48]. As PVR is related to inflammation [10], it is reasonable that IL-10 is suppressed in the retinas of PVR mice and is increased after doxycycline treatment.

With respect to the limitations of our study, previous studies of PVR in mouse models have failed to translate into human patients, likely from multifactorial disease processes not captured in the experimental model. Additionally, mice were treated with IVI doxycycline on the same day when PVR was induced by ARPE-19 cells to ensure the initiation of doxycycline treatment during the early stage of PVR development. In human subjects, we could not start treatment as soon as PVR cells began to form in the vitreous. We also evaluated the efficacy of doxycycline when the drug was applied two days after PVR induction, although the mouse eye samples were too small to receive the second injection. Therefore, other animal models with bigger eyes such as rats or rabbits should be used for further studies to test the effects of doxycycline for PVR in the future.

## 4. Materials and Methods

### 4.1. Cell and Mouse

ARPE-19 cells were purchased from the Bioresource Collection and Research Center (Hsinchu, Taiwan) and were maintained and propagated according to the instructions. C57BL/6J mice were purchased from the National Laboratory Animal Center (Taipei, Taiwan) and bred in our college animal center. All mouse experiments were performed in compliance with a protocol approved by the Institutional Animal Care and Use Committee of National Cheng Kung University (with approval number 105095 on 23 December 2015).

### 4.2. Assays for Cell Migration, Proliferation, and Contraction

Cell migration was assayed via the wound healing assay. Briefly, ARPE-19 cells were seeded into 12-well plates (2 × 10^5^ cells/well), serum-starved for one day, scratched with a 200 μL pipette tip, and incubated in serum-free medium with or without doxycycline (Sigma-Aldrich, Saint Louis, MO, USA). The migration of cells, which closed the wound area, was monitored with a light microscopy and analyzed by ImageJ software. Cell proliferation was assayed by the viability assay. Briefly, ARPE-19 cells were seeded into 96-well plates (5 × 10^3^ cells/100 μL/well) and treated with or without doxycycline. The cell viability was measured using a Multiskan EX microplate reader (Thermo Fisher Scientific, Waltham, MA, USA) after incubating cells with the Cell Counting Kit-8 reagent (Enzo Life Sciences, New York, NY, USA) according to the instructions from the manufacturer. Cell contraction was assayed using the gel contraction assay. Briefly, ARPE-19 cells were mixed with type I collagen from bovine skin (STEMCELL Technologies, Vancouver, BC, Canada), seeded into 24-well plates (3.5 × 10^5^ cells/500 μL/well), then incubated for 1 h for polymerization. The collagen gels were detached from the bottoms of the wells and floated in culture medium with or without doxycycline. The surface area of each collagen gel was monitored using a CCD camera (Thermo Fisher Scientific, Waltham, MA, USA) and analyzed using ImageJ software.

### 4.3. Assays for the Total MMP Activity

ARPE-19 cells were incubated with or without doxycycline for 24 h, harvested, lysed with buffer (Cell Signaling Technology, Danvers, MA, USA), then the concentrations of total proteins in samples were determined using the protein assay dye reagent (Bio-Rad Laboratories, Hercules, CA, USA). Equal amounts of proteins were subjected to determine the total MMP activity and MMP and TIMP levels. The total MMP activity was determined using the MMP Activity Assay Kit (AAT Bioquest, Sunnyvale, CA, USA) according to the instructions from the manufacturer and measured using a Fluoroskan Ascent microplate reader (Thermo Fisher Scientific, Waltham, MA, USA).

### 4.4. Western Blotting Analysis

ARPE-19 cells were processed for the extraction of total proteins via incubation with the cell lysis buffer (Cell Signaling Technology, Danvers, MA, USA), a protease inhibitor cocktail (Sigma-Aldrich, Saint Louis, MO, USA), and PhosSTOP^TM^ (Sigma-Aldrich, Saint Louis, MO, USA). Proteins in samples were measured for concentrations using the protein assay dye reagent. Six mouse retinas were pooled into one sample and processed for the extraction of total proteins via incubation with the cell lysis buffer (Cell Signaling Technology, Danvers, MA, USA) before three 1-min cycles of sonication for 5 s with a pause for 5 s. After removal of insoluble materials via centrifugation with 12,000 rpm for 15 min, the protein concentration of the retinal lysate was measured using the BCA protein assay kit (Thermo Fisher Scientific, Waltham, MA, USA) according to the instructions from the manufacturer. Total proteins in samples were separated via polyacrylamide gel electrophoresis, blotted onto membranes, and blocked with 5% skim milk to prevent nonspecific binding. Blots were stained with the primary antibody against β-actin (Sigma-Aldrich, Saint Louis, MO, USA), human p-p38 MAPK (Cell Signaling Technology, Danvers, MA, USA), human p38 MAPK (Cell Signaling Technology, Danvers, MA, USA), mouse p-p38 MAPK (Cell Signaling Technology, Danvers, MA, USA), or mouse p38 MAPK (Epitomics, Burlingame, CA, USA), followed by secondary antibodies (Jackson Immuno Research Laboratories, West Grove, PA, USA).

### 4.5. PVR Induction and Antibiotic Treatment in Mice

Eight- to nine-week-old female C57BL6/J mice were anesthetized with zeolite and Rompun via intraperitoneal injection. PVR mice were divided into 4 groups based on treatments: (1) saline (*n* = 40); (2) topical tetracycline (*n* = 10); (3) IVI doxycycline (*n* = 14); (4) IVI doxycycline + topical tetracycline (*n* = 35). A hole on the sclera of the mouse eye was poked using a 30G needle and the vitreous humor was removed using a sterile cotton swab. The mouse eyes were intravitreally injected with ARPE-19 (1.6 × 10^4^) cells in Hank’s Balanced Salt Solution with or without doxycycline (5 μg) in 2 μL into the vitreous with a 30G needle on a Hamilton syringe and colloidal, artificial tears or tetracycline ointment were applied topically to avoid liquid outflow. Afterward, topical tetracycline ointment was applied once every other day for 14 days. PVR and inflammation scoring was performed by examining the ocular fundus of mouse eyes with a slit lamp. The severity of PVR was graded on the following scale of 0 to 5: 0, no PVR change; 1, IVI spot or inflammation in the retina; 2, retinal detachment in <25% of the retina; 3, retinal detachment in ≥25% of the retina; 4, ERMs in <25% of the retina; and 5, ERMs in ≥25% of the retina. After PVR induction and doxycycline plus topical tetracycline treatment for 21 days, mouse eyes were harvested and retinas were collected for assays.

### 4.6. Histological and Immunofluorescence Staining

Mouse eyes were collected 21 days after PVR induction and antibiotic treatment then fixed, dehydrated, embedded in paraffin, and sectioned. The sections were stained with H&E. The sections were also subjected to immunofluorescence staining with DAPI (Abcam, Cambridge, UK) for the nucleus and the antibody against mouse MMP-9 (Abcam, Cambridge, UK) or p-p38 MAPK (Cell Signaling Technology, Danvers, MA, USA).

### 4.7. Luminex Assay

Six retinas were pooled into one sample, in which total proteins were extracted and measured as described in Section 4.4. The resulting samples were subjected for the detection of cytokines IL-4, -6, -10, IFN-γ, TNF-α, and vascular endothelial growth factor and chemokines MCP-1, CXCL-9, and CXCL-10 with a Mouse Cytokine/Chemokine Magnetic Bead Panel kit (Merck Millipore, Burlington, MA, USA). Briefly, 25 μL of retinal samples was incubated with antibody-immobilized beads. After a series of wash, detection antibodies were added to the mixture and the cytokines or chemokines were detected by adding streptavidin–phycoerythrin and analyzed with a flow-based Luminex 200 suspension array system (Luminex Corporation, Austin, TX, USA).

### 4.8. Statistical Analyses

Data are expressed as the means ± SEM. For statistical comparison, PVR scores on day 21 post-induction were analyzed using one-way ANOVA followed by the Holm–Sidak test, while the rest of data were analyzed using Student’s *t* test. All analyses were conducted with Prism 7.0 software. Here, *p* values of <0.05 were considered statistically significant, with *, **, and *** indicating *p* < 0.05, <0.01, and <0.001, respectively.

## 5. Conclusions

In this study, we have demonstrated that doxycycline can decrease the migration, proliferation, and contraction of ARPE-19 cells and can suppress MMP activity. Additionally, doxycycline can also ameliorate PVR with reduced MMP-9 expression in mice. These effects are achieved by suppressing p38 MAPK activation in ARPE-19 cells and the mouse retina. Therefore, doxycycline shows promise in inhibiting PVR in mouse models. Future studies are warranted.

## Figures and Tables

**Figure 1 ijms-22-11670-f001:**
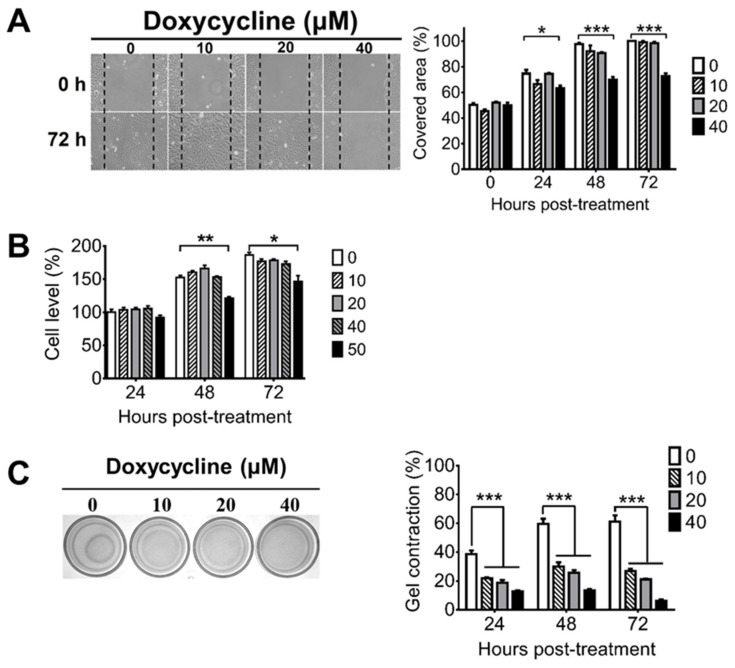
Doxycycline decreases the migration, proliferation, and contraction of ARPE-19 cells. ARPE-19 cells were treated with or without the indicated concentrations of doxycycline and monitored for (**A**) migration, (**B**) proliferation, and (**C**) contraction at the indicated times after drug treatment. The representative samples and quantitated results are shown. The values of ARPE-19 cells without doxycycline treatment for 48 or 24 h are set as 100% in panels A and B, respectively. The data represent means + SEM (error bars) of 3–5 samples per group. Note: *, *p* < 0.05, **, *p* < 0.01, and ***, *p* < 0.001 via Student’s *t* test. In the left panel of C, 72 h samples are shown.

**Figure 2 ijms-22-11670-f002:**
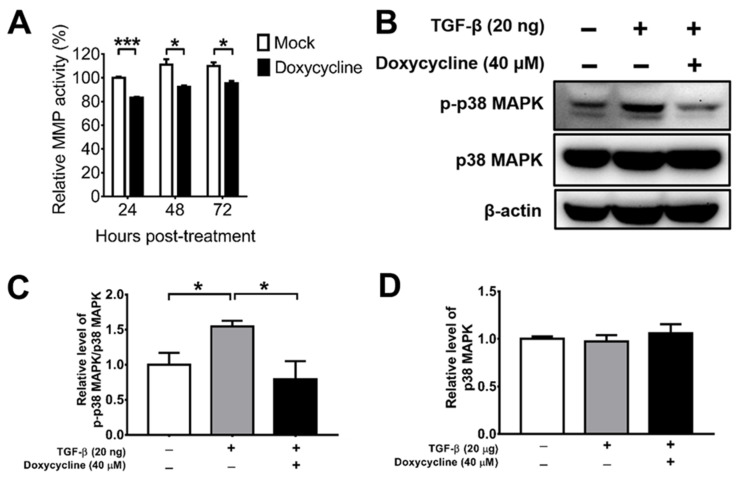
Doxycycline inhibits the total MMP activity and p38 MAPK activation of ARPE-19 cells. (**A**) ARPE-19 cells were treated with or without doxycycline (10 μM) and harvested at the indicated times to assay the total MMP activity. The value of cells without drug treatment for 24 h was set as 100%. (**B**) TGF-β-induced ARPE-19 cells were treated with or without doxycycline (40 μM) for 48 h and assayed for p-p38 MAPK, p38 MAPK, and β-actin via Western blotting. The representative blots (**B**) and quantitated results (**C**,**D**) are shown. In each sample, the values of p-p38 MAPK and p38 MAPK were normalized to that of β-actin and the value of normalized p-p38 MAPK was divided by that of normalized p38 MAPK. The values of cells without TGF-β and doxycycline treatment were set to 1. The data represent means + SEM (error bars) of ≥3 samples per group. Note: *, *p* < 0.05 and ***, *p* < 0.001 via Student’s *t* test.

**Figure 3 ijms-22-11670-f003:**
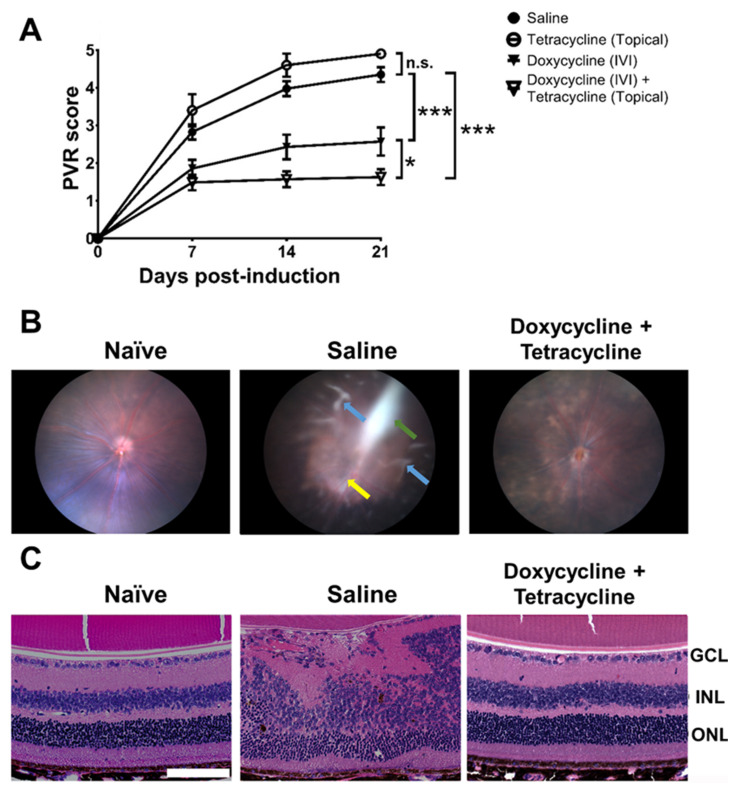
Doxycycline and tetracycline treatment reduces PVR in mice. (**A**) After PVR induction and treatment with saline and doxycycline by IVI injection or tetracycline topically, mice were monitored for disease scores in the fundus using a slit lamp on the indicated days post-induction. The data represent means ± SEM (error bars) of ≥10 mice per data point. Note: *, *p* < 0.05 and ***, *p* < 0.001 via the one-way ANOVA followed by the Holm–Sidak test on day 21 post-induction. Note: n.s.: not significant. (**B**) The representative fundus images of naïve mice or mice with PVR induction and treated with saline or doxycycline and tetracycline with ERM (green arrow), inflammation (yellow arrow), and retinal detachment (blue arrows) are shown. (**C**) The eyes of mice described in (**B**) were harvested 21 days after induction, sectioned, and stained with H&E. GCL: ganglion cell layer; INL: inner nuclear layer; ONL: outer nuclear layer. Scale bar, 100 μm. Data are representative of at least 2 experiments.

**Figure 4 ijms-22-11670-f004:**
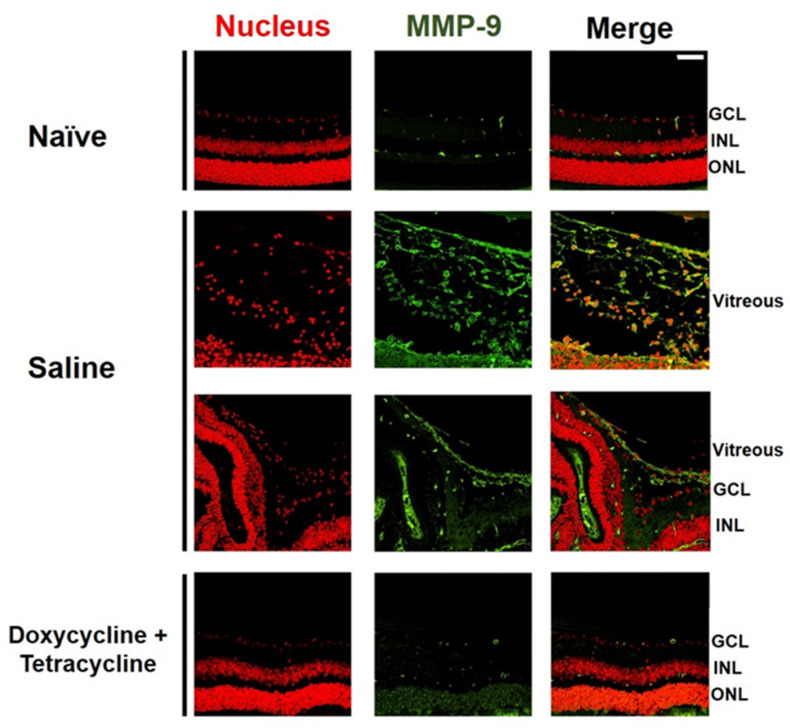
Doxycycline and tetracycline treatment decreases MMP-9 expression in the retina and vitreous of mice after PVR induction. The eye sections of naïve mice or mice with PVR induction and treated with saline or doxycycline and tetracycline were subjected to immunofluorescence staining with DAPI for the nucleus (red) and the antibody against MMP-9 (green) 21 days after induction. GCL: ganglion cell layer; INL: inner nuclear layer; ONL: outer nuclear layer. Scale bar, 50 μm. Data are the representative of at least 2 experiments.

**Figure 5 ijms-22-11670-f005:**
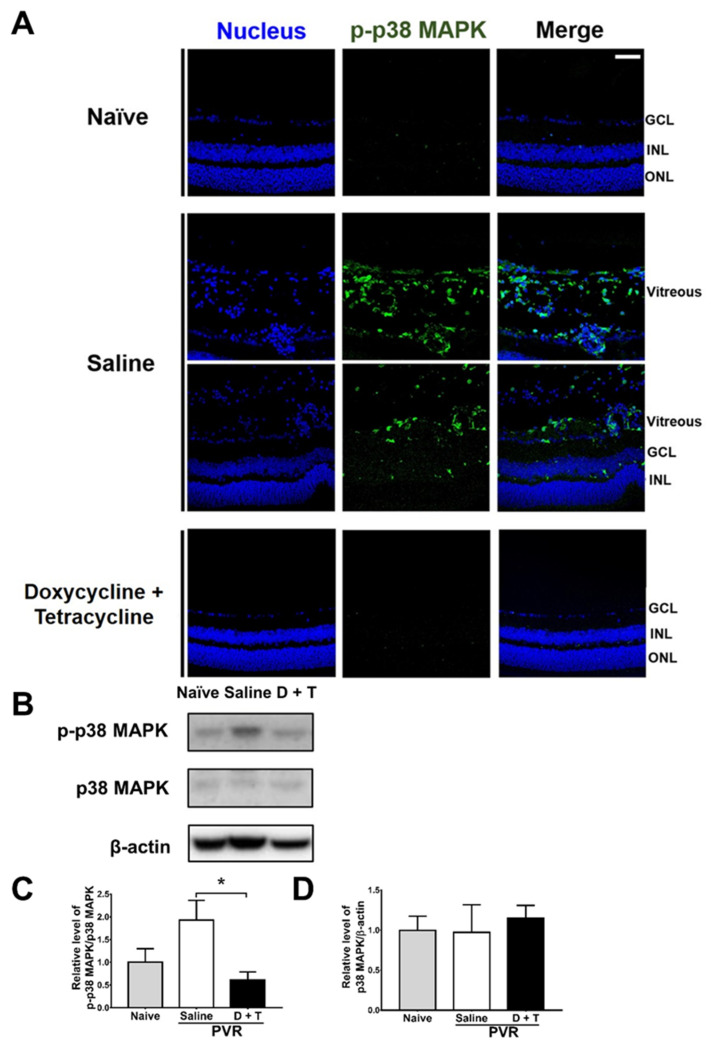
Doxycycline and tetracycline treatment decreases p38 MAPK phosphorylation in the retinas and vitreous of mice after PVR induction. (**A**) The eye sections of naïve mice or mice with PVR induction and treated with saline or doxycycline and tetracycline (D + T) were subjected to immunofluorescence staining with DAPI for the nucleus (blue) and the antibody against p-p38 MAPK (green) 21 days after induction. GCL: ganglion cell layer; INL: inner nuclear layer; ONL: outer nuclear layer. Scale bar, 50 μm. Data are the representative of at least 2 experiments. The retinas harvested from mice as described in (**A**) were subjected to Western blotting for p38 MAPK, p-p38 MAPK, and β-actin. The representative blots (**B**) and quantitated results (**C**,**D**) are shown. In each sample, the value of p-p38 MAPK was divided by that of p38 MAPK. The values of naïve mice were set as 1. The data represent means + SEM (error bars) of ≥3 samples per group. Note: *, *p* < 0.05 via Student’s *t* test.

**Figure 6 ijms-22-11670-f006:**
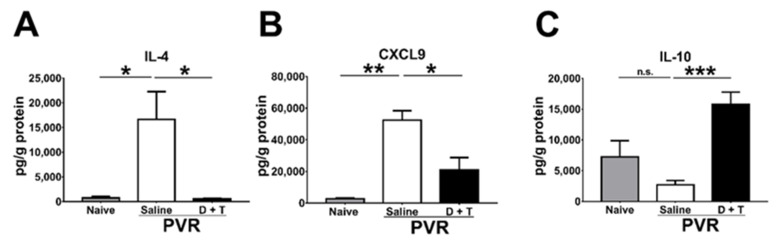
The effects of doxycycline and tetracycline treatment on cytokine and chemokine levels in the retina samples of mice after PVR induction. The retinas harvested from naïve mice or mice with PVR induction and treated with saline or doxycycline plus tetracycline (D + T) were subjected to the Luminex assay to determine the levels of (**A**) IL-4, (**B**) CXCL9, and (**C**) IL-10 21 days after PVR induction. The data represent means + SEM (error bars) of ≥3 samples per group. Note: *, *p* < 0.05, **, *p* < 0.01, and ***, *p* < 0.001 via Student’s *t* test; n.s.: not significant.

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
