# Peer review of "Doxycycline Ameliorates the Severity of Experimental Proliferative Vitreoretinopathy in Mice"

_ijms, 2021, doi:10.3390/ijms222111670_

Round 1

Reviewer 1 Report

The authors have provided in vitro and in vivo evidence that doxycycline can reduce the severty of experimental PVR. In vitro the ARPE-19 cells showed reduced migration, proliferation and contraction when treated with doxycycline, as well as a reduction of p38 MAPK activation and total MMP activity. In their experimental PVR mouse model intravitreal doxycycline decreased levels of IL-4 and MMP-9, which were increased with PVR. Topical doxycycline alone showed no effect on experimental PVR, however together with the intravitreal injections of doxycyline the effect was even stroger.

The topic and the findings of the manuscript are novel and are of clinical interest. The manuscript is structured, well written and understandable. 

The authors introduced a novel mouse model for PVR, and showed convincing in vitro and in vivo data that doxcycycline reduces the severity of PVR.

Reviewer 2 Report

Overall exciting research using a commonly used, safe medication of doxycycline for the prevention of a terrible, blinding disease of PVR. Suggestions for improvement of this manuscript below.

Line 347: RPE cells

Page 2: may also comment that methotrexate is being extensively studied for PVR prevention

Line 265: doxycycline significantly reduces … - use of “really” not scientifically appropriate. Same in line 273

Intravitreal doxy not commercially available. Can oral be used? Comment on ocular penetrance for both doxy and tetracycline.

Methods: state sample size of mouse eyes

Line 407 too strong of conclusion, better to say doxycycline shows promise in inhibiting PVR in this mouse model, future study is warranted.

Limitations section lacking. Previous studies of PVR in mouse models have failed to translate to human patients, likely from multifactorial disease process not captured in the experimental model. One specific limitation to also include is the timing of treatment – in human subjects cannot usually start treatment as soon as PVR cells begin to form in the eye, may be delayed by days to weeks to months – may be too late to see effect from this experimental intervention.

Reviewer 3 Report

In the article by Chen et al., the authors study the potential therapeutic effects of Doxycycline and tetracyline treatment in mouse model of proliferative vitreoretinopathy (PVR). For this, they measure the various molecules that are typically elevated in PVR such as matrix metalloproteinases, cytokines, chemokines etc. and that they were significantly reduced with doxycycline and tetracycline treatment. This study is important from a therapeutic standpoint. However, I have following comments/suggestions on the article.

Major comments-

-Figure 1 and results sub-section 2.1: The difference between different doses 10-40 uM is not explained in the text, were the differences at 24, 48, 72 hours seen at all the doses?

-Figure 1 and results sub-section 2.1: If a significant reduction in proliferation of cells was seen only with 50 uM of Doxycycline, for a proper comparison it would be important to show the effect of 50 uM on migration and contraction as well. 

-Line 110: Please explain the rationale behind introducing a new time point- 36 hours here. If there is nothing interesting about it, please remove it. 72 hours time point however would be useful here, as that would imply how long doxycycline stays effective, in- vitro.

-Line 117: The TIMP levels were checked only at 10 uM and not checked at 40 uM Doxycycline, it’s possible that they might change at 40 uM like MMP8 and MMP9 do. And if the TIMP levels did not change at 40 uM also, please report that.

-Lines 144-145: Please give if there is any reference for the use of tetracycline in this mouse model. It would also be helpful to give a rationale to use a combination of tetracycline and doxycycline, probably in the introduction of the article.

-Lines 147-148: Please mention how much volume of saline was used to inject 5 ug of Doxycycline? Or what was the final molar concentration of Doxycycline injected into mouse eyes?

-Lines 148-149 and Figure 3A: It would be useful to also include the PVR scores of ‘Naive’ retinas here, for readers to be able to compare and fully appreciate the beneficial effects of Doxycycline + tetracycline treatment. 

-Lines 173-174 and Figure 4: From a therapeutic standpoint, it is important to show the effects of doxycycline and tetracycline, before and especially after 21 days. It is possible that the peak effect is at 21 days, which would probably suggest repeating doxycycline and tetracycline at that point.

-Lines 209-212 and Figure S4: At the current magnification, it is hard to conclude from the figure if p65 expression is in nucleus or cytoplasm. So, please provide high magnification images in S4. Otherwise please remove figure S4. completely, the change in nuclear expression of p65 is pretty clear from western blot in figure S3.

-From a therapeutic point-of-view, any potential toxic effects of intravitreal Doxycycline + topical tetracycline on ARPE-19 cell line or in-vivo retina cells in mouse should be reported. For instance, TUNEL assay could be done at 21 days post injection in the mouse retina, and 48-72 hours for ARPE-19 cells and cell death (if any) be compared between treated and untreated (saline) conditions. 

Minor comments-

-Line 55: I think you mean ‘colchicine’ and not ‘chochicine’.

-Line 73: ’Doxycycline has been successfully used in the clinical acne treatment..’ The use of doxycycline for acne treatment is not relevant to this study and should be removed.
